



# Dimethylated sulfur compounds in the Peruvian upwelling system

Yanan Zhao[1], Dennis Booge[1], Christa A. Marandino[1], Cathleen Schlundt[1], Astrid Bracher[2,3], Elliot L. Atlas[4], Jonathan Williams[5,6] and Hermann W. Bange[1]

[1]GEOMAR Helmholtz Centre for Ocean Research Kiel, Kiel, Germany
[2]Phytooptics Group, Physical Oceanography of Polar Seas, Climate Sciences, Alfred Wegener Institute, Helmholtz Centre for Polar and Marine Research, Bremerhaven, Germany
[3]Department of Physics and Electrical Engineering, Institute of Environmental Physics, University of Bremen, Bremen, Germany
[4]Rosenstiel School of Marine and Atmospheric Science, University of Miami, Miami, Florida, USA
[5]Atmospheric Chemistry and Multiphase Chemistry Departments, Max Planck Institute for Chemistry, 55128 Mainz, Germany
[6]Energy, Environment and Water Research Centre, The Cyprus Institute, 1645 Nicosia, Cyprus

*Correspondence to:* Y. Zhao (yzhao@geomar.de)

**Abstract.** Our understanding of the biogeochemical cycling of the climate-relevant trace gas dimethylsulfide (DMS) in the Peruvian upwelling system is still limited. Here we present, oceanic and atmospheric DMS measurements which were made during two shipborne cruises in December 2012 (M91) and October 2015 (SO243) in the Peruvian upwelling region. Dimethylsulfoniopropionate (DMSP) and dimethylsulfoxide (DMSO) were also measured during M91. Relatively low DMS concentrations were measured in surface waters in October 2015 ($1.9 \pm 0.9$ nmol L$^{-1}$) and December 2012 ($2.5 \pm 1.9$ nmol L$^{-1}$). Nutrient availability appeared to be the main driver of the observed variability in the surface DMS distributions in the coastal areas. DMS, DMSP and DMSO showed maxima in the surface layer and no elevated concentrations associated with the oxygen minimum zone off Peru were measured. The possible role of DMS, DMSP and DMSO as radical scavengers (stimulated by nitrogen limitation) is supported by their negative correlations with N:P (sum of nitrate and nitrite: dissolved phosphate) ratios. Large variations in atmospheric DMS mole fractions were measured during M91 ($144.6 \pm 95.0$ ppt) and SO243 ($91.4 \pm 55.8$ ppt); however, the atmospheric mole fractions were generally low, and the sea-to-air flux density was primarily driven by seawater DMS. The Peruvian upwelling region was identified as a source of atmospheric DMS in December 2012 and October 2015, however, in comparison to the global monthly Lana climatology (mean: 6.2–9.8 µmol m$^{-2}$ d$^{-1}$ in October/December) (Lana et al., 2011), the Peru upwelling was not a hotspot of DMS emissions at either time (M91: $5.9 \pm 5.3$ µmol m$^{-2}$ d$^{-1}$; SO243: $3.8 \pm 2.7$ µmol m$^{-2}$ d$^{-1}$).

## 1 Introduction

The trace gas dimethylsulfide (DMS) is mainly produced in the marine environment and serves as one of the most abundant biogenic atmospheric sulfur sources, transferring approximately 28.1 Tg sulfur from the oceans into the atmosphere annually as estimated by Lana et al. (Lana et al., 2011). DMS can be oxidised to form sulfate aerosols in the atmosphere which, in turn,



play an important role in regulating Earth's climate via altering cloud properties and modulating cloud albedo (Stefels et al., 2007). However, the global significance of this DMS-climate link is still under debate (Quinn and Bates, 2011).

Oceanic DMS is mainly produced from its precursor dimethylsulfoniopropionate (DMSP) which is synthesised in phytoplankton cells and then released into the seawater by senescence, virus attacks, and grazing (Stefels et al., 2007). Cellular DMSP concentrations vary in different phytoplankton groups: dinoflagellates and prymnesiophytes are major DMSP producers, whereas diatoms produce less DMSP (Keller, 1989). DMSP can be degraded to methanethiol via demethylation (Howard et al., 2006) or to DMS and acrylate via lyase-mediated cleavage (Curson et al., 2011). The conversion of DMSP to DMS (i.e.,

cleavage pathway) is only of minor importance for DMSP loss, as up to 75 % of DMSP is metabolised by marine bacterioplankton through the demethylation pathway (Moran et al., 2012).

The formation and consumption processes of dimethylsulfoxide (DMSO) remain less clear, but DMSO is known to be a photochemical and biological oxidation product of DMS (Hatton et al., 2012). Some studies showed that it could be directly synthesised in marine phytoplankton cells (Lee et al., 1999). A recent study suggested that dissimilation of dissolved DMSO

($DMSO_d$) to carbon dioxide can be a significant loss pathway in coastal waters (Dixon et al., 2020). Both DMSP and DMSO exert similar intracellular functions (e.g., cryoprotection and antioxidation) in phytoplankton cells (Simó et al., 1998; Sunda et al., 2002) and can support bacterial carbon and sulfur demand (Simó et al., 2002; Dixon et al., 2020).

Biologically productive regions of the ocean can be a significant source of DMS to the atmosphere (Lana et al., 2011). The eastern tropical South Pacific Ocean (ETSP) features one of the four major Eastern Boundary Upwelling Ecosystems (EBUS):

year-round coastal upwelling off Peru and northern Chile is driven by offshore Ekman transport resulting from alongshore trade winds, which leads to abundant nutrients in surface waters and enhanced biological production (Chavez and Messié, 2009). The Peru upwelling system is significantly influenced by El Niño-Southern Oscillation (ENSO) events at decadal scales (Espinoza-Morriberón et al., 2017). ENSO is a natural climate event that affects the oceanic and atmospheric conditions worldwide and occurs in intervals between two and ten years (Santoso et al., 2017; Timmermann et al., 2018). ENSO is divided

into three phases: El Niño (the warm phase), La Niña (the cold phase), and the neutral phase. El Niño/La Niña events are characterised by warming/cooling of sea surface temperature (SST) in the equatorial Pacific Ocean and result in the deepening/shoaling of thermocline off Peru (Dewitte et al., 2012). During El Niño, the thermocline is pushed away from the surface layer (i.e., reduced coastal upwelling), resulting in limited nutrient supply and leading to a decline in primary production in the ecosystem (Barber and Chavez, 1983). Although upwelling of cold and nutrient-rich waters can be observed

along the coast of Peru year-round (Tarazona and Arntz, 2001), surface chlorophyll concentration peaks in austral summer and decreases in austral winter, which is out of phase with the upwelling intensity (Chavez and Messié, 2009). This paradoxical seasonal cycle may arise from both light limitation and dilution due to the deepening of the mixed layer in winter (Echevin et al., 2008).

The Peru upwelling system has been reported to be a source of biogenic halocarbons (e.g., bromoform and methyl iodide) both

in surface seawater and atmosphere (Hepach et al., 2016); however, the measurements of biogenic sulfur compounds (DMS,





DMSP and DMSO) are relatively sparse in this area (Andreae, 1985; Riseman and DiTullio, 2004; Yang et al., 2011). As a result, abiotic and biotic factors affecting DMS/P/O cycling and distributions are not fully understood. We measured DMS concentrations in the water column in the Peru upwelling region as well as atmospheric DMS mole fractions during two research campaigns in December 2012 (M91) and October 2015 (SO243). In addition to DMS, particulate and dissolved DMSP

and DMSO concentrations were measured during M91. This study is intended to focus on (i) deciphering the distributions of DMS, DMSP, and DMSO in the Peru upwelling, (ii) identifying their main drivers and (iii) quantifying the DMS emissions to the atmosphere in order to re-assess the role of DMS emissions from the Peru upwelling region.

## 2 Material and Methods

### 2.1 Sampling sites

The cruise M91 (Fig. 1a) was conducted on board the R/V Meteor between 1 and 26 December 2012. The cruise started from the northernmost location at 5 °S and moved to the southernmost position at 16.2 °S with several transects perpendicular to the coastline of Peru. Underway samples were taken from a continuously operating pump in the ship's hydrographic shaft (at ~ 7 m water depth), and profiles were made between 3 and 2000 m water depth from a 24 × 12 L Niskin bottle rosette equipped with a probe for conductivity, temperature and water depth (CTD). An overview of the methods used for determining

oceanographic parameters such as oxygen and nutrients can be found in Czeschel et al. (2015). The cruise SO243 (Fig. 1b) took place on board the R/V Sonne between 5 and 22 October 2015, from Guayaquil, Ecuador to Antofagasta, Chile, with perpendicular transects to the shelf between ~ 9–16 °S (Fig. 1b). Underway samples were taken from a continuously operating pump in the ship's hydrographic shaft (at ~ 5 m water depth), and profile samples were made between 5 and 150 m from 24 × 10 L Niskin bottle rosette equipped with a CTD. Measurements of oceanographic parameters such as oxygen and nutrients

are described in Stramma et al. (2016). Generally, both cruises alternated between offshore and onshore transects (Fig. 1) in the Peru upwelling region (4–16 ºS). Therefore, the sampling stations of M91 and SO243 can be categorised into offshore and coastal stations according to their respective depths: stations shallower than 300 m are defined as coastal stations and those deeper than 1000 m are defined as offshore stations. In addition to underway samples, seawater column-integrated (1–10 m) samples from the CTD casts are defined as surface samples for M91.

The Ocean Niño index (ONI; http://origin.cpc.ncep.noaa.gov/products/analysis_monitoring/ensostuff/ONI_v5.php), which quantifies the progress and strength of El Niño, suggests that M91 (ONI < 0.5 °C from August 2012 to March 2013) took place in a neutral phase while SO243 occurred during a strong El Niño event (ONI ≥ 0.5 °C from November 2014 to May 2016) (Santoso et al., 2017). However, the El Niño during SO243 was still developing with an ONI of 2.2 °C for Aug–Oct 2015.





## 2.2 Sulfur compounds measurements

Sulfur compounds (DMS/P/O) on M91 were analysed by purge and trap coupled to a gas chromatograph-flame photometric detector (GC-FPD) as described in Zindler et al. (2012, 2013). Briefly, DMS samples were analysed immediately after sampling and filtering (GF/F; Whatman; 0.7 µm) during the cruise, while DMSP and DMSO samples were analysed after returning to the lab. Dissolved DMSP (DMSP$_d$) samples were measured after adding sodium hydroxide (NaOH; Carl Roth) to the DMS samples, and total DMSP (DMSP$_t$) was measured from the unfiltered alkaline samples. Particulate DMSP (DMSP$_p$)

concentrations were calculated by subtracting measured DMS and DMSP$_d$ concentrations from measured DMSP$_t$ concentrations. Dissolved DMSO (DMSO$_d$) and total DMSO (DMSO$_t$) samples were measured from the same samples of DMSP$_d$ and DMSP$_t$ measurements by adding cobalt-dosed sodium borohydride (NaBH4; Sigma-Aldrich) after DMSP measurements. Particulate DMSO (DMSO$_p$) concentrations were calculated by subtracting measured DMSO$_d$ concentrations from measured DMSO$_t$ concentrations. The mean precision of all sulfur compound measurements using GC-FPD was ± 18 %.

Seawater DMS on SO243 was measured using a purge and trap system attached to a gas chromatograph-mass spectrometer (GC-MS), as described in Zavarsky et al. (2018). The mean precision of the DMS measurements using GC-MS was ± 11 %. Dissolved DMS data measured simultaneously using GC-FPD and GC-MS at the GEOMAR time-series stations Boknis Eck (BE; www.bokniseck.de) are used to quality control the comparison of the M91 and SO243 datasets. The BE measurements are performed at 6 different depths on a monthly schedule and an assessment of a year-long dataset shows good agreement

(Fig. S1). The BE relationship was used to adjust the dissolved DMS data from M91, which resulted in an increased uncertainty of ± 30.6%. DMS atmospheric mole fractions during both cruises were measured from whole air samples collected in 2.3 L electropolished stainless steel canisters. Sample analysis was performed on a multi-channel GC/MS/FID/ECD system (Agilent 7890 GC, 5973 MS), which used a Markes Unity system for sample concentration (Andrews et al., 2016). Samples were collected approximately every 3 hours (in parallel with seawater DMS samples). In addition, measurements of DMS

atmospheric mole fraction (parts per trillion) during M91 were performed using a commercial proton-transfer-reaction time-of-flight mass spectrometer (PTR-TOF-MS) from Ionicon Analytik GmbH (Innsbruck, Austria). Continuous mass spectra were collected ranging from m/z 10–350 and averaged such that the time resolution of the measurements was set to 30 seconds. The mean precision of the atmospheric DMS measurements using PTR-TOF-MS was ± 18 %. The comparison between atmospheric DMS data using two methods during M91 shows a good correlation (Fig. S1), but not 1:1 agreement. It is not

possible to determine which dataset is most accurate. In this study, we used the atmospheric DMS PTR-TOF-MS data which were adjusted to the discrete canister samples because of the higher temporal resolution of the continuous PTR-TOF-MS measurements, which resulted in an increased uncertainty of ± 24 %.

## 2.3 Calculation of sea-to-air flux densities

Sea-to-air flux densities, F (µmol m$^{-2}$ day$^{-1}$), of DMS were calculated according to Eq. (1):

$$F = k_w(C_w - x'P/H), \tag{1}$$





where $C_w$ and x' are the dissolved DMS surface concentrations and the DMS atmospheric dry mole fractions, respectively. P is the ambient pressure (set to 1 atm) and H is Henry's law solubility coefficient (Dacey et al., 1984), $k_w$ is the gas transfer velocity calculated with the in-situ wind speed (U) of the shipboard observations and a Schmidt number (Sc) normalised to 600 (which was chosen to allow a direct comparison to the Lana et al. (2011) climatology), according to Nightingale et al.
(2000); thus $k_w$ was calculated according to Eq. (2):

$$k_w = (0.222U^2 + 0.33U)(Sc/600)^{-0.5}, \tag{2}$$

where Sc was calculated using SST according to Saltzman et al. (1993) and the wind speed measurements were corrected to a height of 10 m above the sea level following the method of Hsu et al. (1994).

## 2.4 Phytoplankton pigments and Chlorophyll *a* concentration of major groups

Phytoplankton pigment samples were generally taken in parallel with DMS samples in depths between ~ 3–200 m during both cruises. The process for measuring phytoplankton pigments and further converting them into phytoplankton groups are identical for the cruises M91 and SO243. Pigment samples were filtered through Whatman GF/F filters at the stations where DMS was sampled and then stored at -80 °C until analysis. Pigment concentrations, as described in Booge et al. (2018), were determined using high-pressure liquid chromatography (HPLC) according to the method of Barlow et al. (1997), which was
adjusted to our temperature-controlled instruments as detailed in Taylor et al. (2011). Pigments, as listed in Table 2 of Taylor et al. (2011), were quality controlled according to Aiken et al. (2009), and were published in Hepach et al. (2016) and Bracher (2019). Phytoplankton composition was derived, as in Booge et al. (2018) already for SO243 also for M91 data: Using the diagnostic pigment analysis developed by Vidussi et al. (2001) and subsequently refined by Uitz et al. (2006), seven marker (diagnostic) pigments as proxies for specific phytoplankton groups (diatoms, dinoflagellates, haptophytes, chrysophytes,
cryptophytes, cyanobacteria (excluding prochlorophytes), and chlorophytes) are used to relate their weighted sum of concentrations to the sum of monovinyl-Chl *a*. By applying then the specific weights determined by Uitz et al. (2006) the specific Chl a for each group is determined. The Chl *a* of prochlorophytes was directly derived from the divinyl-Chl *a* (the marker pigment for this group). The eight identified phytoplankton groups are assumed to represent the entire phytoplankton community and the sum of their Chl *a* represents the total Chl *a*.

## 3 Results and Discussion

### 3.1 Cruise settings

M91 took place in the coastal, wind-driven Peruvian upwelling system with southeasterly winds at $6.6 \pm 2.1$ m s⁻¹. The mean $\pm$ std (min–max) SST and sea surface salinity (SSS) during M91 were $18.9 \pm 2.1$ (15.0–22.4) °C and $35.1 \pm 0.1$ (34.9–35.3), respectively. Generally, SSTs below 18.0 °C were measured at the coastal stations (Fig. 2a), where upwelled water was found.
The mean Chlorophyll *a* (Chl *a*) concentration was $2.5 \pm 2.3$ (0.3–7.9) µg L⁻¹, displaying a decreasing trend in the offshore





direction (Fig. 2a). Diatoms were the most dominant algae group and were observed at all stations in surface waters with the mean relative abundance of 41 %, followed by haptophytes (24 %) and chlorophytes (10 %) in the phytoplankton community (Hepach et al., 2016). The N:P ratio, defined as the ratio of the sum of nitrate ($NO_3^-$) and nitrite ($NO_2^-$) to dissolved phosphate ($PO_4^{3-}$) for both cruises, is a good indicator of nutritional status: high/low N:P ratios indicate nitrogen repletion/limitation. N:P

ratios ranged between 0.1 and 13.9 during M91, with a general decreasing trend southward and nitrogen-deficient conditions at coastal stations near Callao (F1–F4; Fig. 2a), which may suggest different phytoplankton bloom stages (e.g., a fully developed bloom at station F1 and F2 as indicated by higher Chl a and lower N:P values).

SO243 started with a section passing the Equator at around 85.5 °W with a mean (± std) SST and SSS of 25.2 ± 1.5 °C and 34.2 ± 0.7 as well as low Chl $a$ values 0.4 ± 0.2 µg L$^{-1}$ (Fig. 2b). Afterwards, SST, SSS, and Chl $a$ values between stations 5

and 18 were 18.8 ± 1.6 (15.6–20.5) °C, 35.2 ± 0.1 (35–35.4), and 2.5 ± 2.1 (0.5–7.2; Fig. 2b) µg L$^{-1}$, respectively, with constant southeasterly winds of 8.2 ± 2.5 m s$^{-1}$, comparable to those measured during M91. The phytoplankton composition during SO243 is similar to M91, with the most abundant phytoplankton groups being diatoms (45 %), haptophytes (24 %), and chlorophytes (18 %) (2018). N:P ratios were generally between 8–13 in the Peru upwelling region during SO243, indicating slightly limiting nitrogen conditions (Fig. 2b). Although the cruise SO243 took place during the strong El Niño event in

2015/2016, typical El Niño signals of warmer, saline, and oxygen-depleted upwelled waters were only apparent at the shelf stations at ~ 9 °S off Peru. The El Niño signals were still weak at ~ 12, 14, and 16 °S transects as regularly cold and nutrient-rich upwelled waters were still present at the easternmost stations of these transects (Stramma et al., 2016). Overall, this indicates that at the time of our measurements in October 2015, the El Niño signal was not fully developed.

## 175  3.2 Seawater DMSP and DMSO

During M91, the mean concentrations of DMSP$_p$, DMSP$_d$, DMSO$_p$, and DMSO$_d$ in surface waters were 85.0 ± 68.9 (10.3–288.9) nmol L$^{-1}$, 16.8 ± 16.5 (0.8–61.9) nmol L$^{-1}$, 40.1 ± 45.4 (2.8–167.9) nmol L$^{-1}$ and 14.0 ± 9.5 (1.3–28.9) nmol L$^{-1}$, with DMSP$_p$ and DMSO$_p$ displaying a decreasing trend in an offshore direction along the F transect parallel to Callao (Fig. 3, a–d). Vertical profiles at station F4 exhibited typical patterns of these sulfur compounds and Chl $a$ during M91 (Supporting

Information; Fig. S2): higher concentrations were measured in surface waters than those in bottom waters, with no enhanced signals in the oxygen minimum zone (OMZ).

The significant correlation between DMSP$_p$ and DMSO$_p$ (DMSO$_p$ = 0.62 * DMSP$_p$ – 0.91, $r^2$ = 0.8; see also Table 1) in this study is in good agreement with a correlation reported previously ([DMSO$_p$] = 0.73 × [DMSP$_p$] – 0.66, $r^2$ = 0.9) off the Peruvian coast in September 2000 (Riseman and DiTullio, 2004). The strong correlation and remarkable similarity of the ratio

between the two particulate sulfur compounds from these two studies spanning a decade interval may indicate that they share a similar biogenic source and/or have related physiological functions in phytoplankton cells (Stefels et al., 2007). However, the concentrations of DMSP$_p$ (1–45.7 nmol L$^{-1}$) and DMSO$_p$ (0.1–30.8 nmol L$^{-1}$) reported by Riseman & DiTullio (2004) are





generally lower than those measured in our study. Considering phytoplankton community composition (dominated by diatoms) and biomass (represented by Chl *a*) are comparable between the two studies, and no significant relationship was observed

between individual phytoplankton groups and DMSP(O)$_p$ during M91 (Table 1), phytoplankton does not seem to be the main driver for the difference of DMSP(O)$_p$ between the two studies. In contrast, nutrient availability (indicated by N:P ratios) might be the reason. Sunda *et al.* (2007) reported a substantial increase in intracellular DMSP concentrations in coastal diatoms under nitrogen limitation, which is assumed to be in response to oxidative stress within phytoplankton cells. Recently, Theseira et al. (2020) observed the accumulation of DMSP in diatoms (*Thalassiosira weissflogii*), which can reduce intracellular reactive

oxygen species. N:P ratios were generally low (< 10) during M91 while they were > 10 (NO$_x$ > 17 µmol L$^{-1}$, PO$_4^{3-}$ > 1.2 µmol L$^{-1}$) in September 2000 (Riseman and DiTullio, 2004). Therefore, higher DMSP$_p$ and DMSO$_p$ concentrations during M91 were potentially up-regulated in response to increased oxidative stress induced by nutrient limitation, and this is in line with negative correlations between N:P ratios and DMSP(O)$_p$ (Table 1). Moreover, negative correlations (Table 1) between ß-carotene (a known antioxidant) and DMSP(O)$_p$ concentrations in both studies further support this speculation. Additionally, Riseman &

DiTullio (2004) reported enhanced concentrations of DMSP(O)$_p$ under low-iron conditions, which exemplifies that either macro-, micronutrients or co-limitation conditions might elevate DMSP(O)$_p$ concentrations against oxidative stress.

In contrast to our observations, Zindler et al. (2012) reported a general decreasing trend of DMSP$_t$ concentrations with decreasing N:P ratios (1–12). This may be because the response to nitrogen limitation differs among specific algae groups. For instance, Gaul (2004) reported decreasing DMSP production rates with decreasing nitrogen concentration among nine strains

within the groups of dinoflagellates, diatoms, prymnesiophytes and cryptophytes. Therefore, despite the fact that diatoms were the dominant algae groups off Peru and off Mauritania, variability at the species or genus level might result in different responses under nitrogen limitation.

### 3.3 Seawater DMS

Surface DMS concentrations were variable during M91, with a mean concentration of 2.5 ± 1.9 (0.2–8.2) nmol L$^{-1}$ (Fig. 3e).

Elevated concentrations were observed at the major upwelling centres (e.g., Chimbote and Callao), with the highest value measured north of Chimbote. The cross-shelf section starting from Callao during M91 displayed decreasing trends of DMS towards an offshore direction. DMS concentrations of 1.9 ± 0.9 (0.5–4.5) nmol L$^{-1}$ (Fig. 3f) in surface waters during SO243 could be grouped into three sections: the equatorial section (Guayaquil–station 5), the Peruvian upwelling section (stations 6–18) and the Chilean waters section, with mean concentrations of 2.4 ± 0.8 nmol L$^{-1}$, 1.2 ± 0.8 nmol L$^{-1}$ and 2.1 ± 0.7 nmol L$^{-1}$,

respectively. DMS values were generally low in the Peru upwelling section, regardless of coastal or offshore areas. Similar to DMSP and DMSO, vertical profiles of DMS for both cruises presented decreasing trends with increasing depths (Fig. S2) and again, no elevated concentrations were measured associated with OMZ in the water column, which is in agreement with Andreae (1985).





Except for the positive correlation between DMS and $DMSO_d$ during M91 (Table 1), which serves as a hint that DMS photo-

degradation was an important source of $DMSO_d$ in surface waters (Xu et al., 2021), DMS was significantly correlated only
with N:P ratios for both cruises (Table 1). Nutrient availability is essential to plankton (phytoplankton and bacterioplankton)
communities and therefore, N:P ratios, as well as fixed nitrogen deficit (Ndef), were used as indirect indicators to investigate
the nutrient status on the coastal DMS distributions for both cruises. Ndef was calculated using the following equation (Graco
et al., 2017):

$$Ndef = 12.6 \times [PO_4^{3-}] - [NO_3^-] - [NO_2^-], \quad (3)$$

where the constant 12.6 is the empirical N:P ratio of organic matter produced in the ETSP (Codispoti and Packard, 1980), and
positive Ndef indicates nitrogen-depleted conditions. Generally, both N:P ratios and Ndef significantly correlated with coastal
DMS values in the surface waters (Fig. 4). Enhanced DMS concentrations associated with nitrogen limitation could be linked
to an increase in the activity of the DMSP lyase enzyme, probably in response to increased oxidative stress (Sunda et al., 2007).

This might also explain the discrepancy in coastal DMS values between the two cruises: higher DMS values from M91 might
be up-regulated in a more nitrogen-limited environment, as compared to SO243. Our observation of nutrient availability and
DMS concentrations are in line with other field studies (Leck et al., 1990; Zindler et al., 2012).

### 3.4 Seawater DMS comparison

DMS is heavily undersampled in ETSP, especially for the coastal areas (defined as bathymetry < 300 m in this study), with

only one other published DMS dataset from June–July 1982 (Andreae, 1985). Coastal DMS concentrations reported by
Andreae (1985) are $6.5 \pm 6.2$ (1–43.9) nmol $L^{-1}$ (Fig. 3e) which is much higher compared to the measurements reported here.
However, Andreae (1985) reported comparable mean Chl *a* values (dominated by diatoms) and intermediate N:P ratios (9–
11). Therefore, phytoplankton biomass or nitrogen limitation would not explain the extremely high DMS concentrations
measured by Andreae (1985). One possible reason could be the seasonality of the upwelling strength, as upwelling is strongest

in austral winter (Echevin et al., 2008) and more intense upwelling might be favourable to the biogenic production of DMS.
This is in line with the highest DMS concentrations measured at the southern upwelling centres between Chimbote and Callao
(> 40 nmol $L^{-1}$), where the most intense upwelling occurred (Andreae, 1985).

To examine our data in a broader context, DMS data from the global Surface Seawater DMS database (PMEL;
http://saga.pmel.noaa.gov/dms) and DMS monthly climatology from Lana et al. (2011) have been extracted. Generally, DMS

concentrations reported in our study are comparable to those from the PMEL database (Fig. 3f) (Hind et al., 2011) and the
Lana climatology (background in Fig. 3e and 3f) in open ocean ETSP, which is close to those published in the equatorial
Pacific (Bates and Quinn, 1997). Only DMS concentration data in October–November 2007 (triangles in Fig. 3f) are slighter
lower in adjacent waters, which were more likely to be affected by advective processing from the coastal environments,
especially considering that the PMEL 2007 cruise took place in a La Niña event indicated by ONI (ONI < -0.5 between July





2007 and July 2008), which likely triggers stronger westward propagation (Hu et al., 2014). Indeed, we measured lower DMS concentrations in the coastal region off Peru during SO243.

To investigate the effects of ENSO events (represented by the ONI) on surface DMS concentrations in the ETSP, more DMS datasets are included, which covers the last four decades (Hatton et al., 1998; Huebert et al., 2004; Turner et al., 1996). The result indicates that no relationship is found between ONI and surface DMS concentrations in the ETSP (Fig. 5a), which is in

agreement with the finding of a previous study in the equatorial Pacific Ocean (Bates and Quinn, 1997). A comparison with DMS data from other EBUS and the Arabian Sea illustrates that DMS concentrations off Peru (up to 44 nmol L$^{-1}$) are higher than those measured off northwest Africa (Mauritania and Morocco) (Belviso et al., 2003; Franklin et al., 2009; Zindler et al., 2012), Benguela (Andreae et al., 1994), California (Herr et al., 2019) and the west Arabian Sea (Oman) (Hatton et al., 1999). Only DMS concentrations reported from the east Arabian Sea (Shenoy and Kumar, 2007) are comparable or much higher

during non-upwelling or upwelling conditions, respectively (Fig. 5b). However, compared to global coastal DMS distributions, DMS concentrations in the Peru upwelling are in the average range (see Table 2 in Zhao et al., 2021), especially in October and December.

### 3.5 DMS flux densities

To our knowledge, this is the first report of atmospheric DMS distributions above the coastal Peruvian upwelling. We used

the Hybrid Single-Particle Lagrangian Integrated Trajectory (HYSPLIT; http://www.arl.noaa.gov/HYSPLIT.php) model to calculate air mass backward trajectories (24 h; starting height: 50 m) based on the DMS sampling sites for both cruises. The trajectories convey that the air masses encountered on both cruises were from the ocean for more than 24 h (Fig. 6a and 6b) and 48 h (figures not shown) prior to sampling and thus, the air masses sampled during the two campaigns most likely had not been affected by terrestrial DMS sources.

DMS mole fractions ranged from below the detection limit (~ 25 ppt) to 571.3 (mean: 144.6 ± 95.0) ppt during M91 (Fig. 7a), with the highest value measured around Pisco. Generally, DMS mole fractions did not follow the pattern of dissolved DMS in surface seawater during M91; however, were affected by the wind speeds, which could affect the sea-to-air exchange process. This was supported by the positive correlation between atmospheric DMS mole fractions and wind speeds ($r = 0.39$, $p < 0.01$, n = 44). DMS mole fractions varied between 11.6 and 253.3 (mean: 91.4 ± 55.8) ppt during SO243 (Fig. 7b), with the same

distribution as dissolved DMS in the surface layer ($r = 0.43$, $p < 0.01$, n = 96;). Generally, atmospheric DMS mole fractions measured in the open ocean ETSP (~ 100 ppt) during SO243 were within the same magnitude as those measured by Huebert et al. (2004) (~ 150 ppt) between 7.5 °N–7.5 °S over the equatorial Pacific Ocean. However, atmospheric DMS mole fractions measured over the coastal Peru upwelling were lower (~ 65 ppt) compared to those measured in similar regions during M91 (~200 ppt), which might be a result of the discrepancy for DMS flux densities between the two cruises (Fig. 7). Overall, the

mean atmospheric DMS mole fractions measured during M91 and SO243 were comparable and they fell within the range between 61–340 ppt previously reported over the South Pacific Ocean (Lee et al., 2010; Marandino et al., 2009).



DMS flux densities ranged from 0.4 to 28.2 (mean: 5.9 ± 5.3) µmol m$^{-2}$ d$^{-1}$ during M91 (Fig. 7a), with the highest value at station J1. Although a few coastal sampling sites where low wind speeds were associated with high seawater DMS concentrations  (e.g., Callao), DMS flux densities generally showed a similar distribution as the seawater DMS distribution ($r$

= 0.68, $p$ < 0.01, n = 44) during M91. DMS flux densities during SO243 ranged between 0.3 and 30.5 (mean: 7.4 ± 5.4) µmol m$^{-2}$ d$^{-1}$ (Fig. 7b), with the peak DMS flux density around the equator. DMS flux densities were mainly driven by seawater DMS concentrations ($r$ = 0.68, $p$ < 0.01, n = 93) and exhibited similar distributions: 10 ± 5.8 µmol m$^{-2}$ d$^{-1}$ in the equatorial section, 3.8 ± 2.7 µmol m$^{-2}$ d$^{-1}$ in the Peru upwelling section and 6.5 ± 3.7 µmol m$^{-2}$ d$^{-1}$ in the Chilean waters section.

On average, the mean DMS flux densities in the first and third regions during SO243 are slightly higher/comparable to 5.5

µmol m$^{-2}$ d$^{-1}$ reported from Marandino et al. (2009) and 7.2 µmol m$^{-2}$ d$^{-1}$ from Omori et al. (2017) in the Southern Pacific Ocean. The lower DMS flux densities in the second region during SO243 (the coastal Peru upwelling) are generally comparable to those reported from Yang et al. (2011) in northern coastal Chile (~ 0–4 µmol m$^{-2}$ d$^{-1}$) and are much lower than those obtained from M91 in similar regions, which is primarily driven by the discrepancy in seawater DMS concentration. In general, DMS flux densities obtained from M91 and SO243 over the coastal Peruvian upwelling are lower compared to the monthly (October

and December) global (coastal and oceanic regions) mean DMS flux densities (6.2–9.8 µmol m$^{-2}$ d$^{-1}$) estimated by Lana et al. (2011), revealing that Peruvian upwelling is not a significant source of DMS to the atmosphere during both survey periods. In addition, the difference of DMS flux density calculation with and without considering the atmospheric DMS mixing ratios indicates DMS flux densities (calculated only by seawater DMS) were not greatly overestimated (< 10 %) in the Peru upwelling region.

## 4. Summary


As one of the world's most productive oceanic regions,  the upwelling region off Peru is of great interest for studying biogenic trace gas production and its emissions to the atmosphere. We present here, for the first time, simultaneously measured DMS/P/O seawater concentrations and DMS atmospheric mole fractions from the Peruvian upwelling region during two cruises in December 2012 and October 2015. Large variations were determined in seawater DMS concentrations off Peru

upwelling. Anticorrelations were found between nutrient availability (represented by the N:P ratios) and sulphur compounds, which may reinforce their potential roles as antioxidants in response to oxidative stress. We found a significant relationship between DMS concentrations and nitrogen availability related indicators in the coastal region, pointing to the importance of the interaction between environmental parameters and DMS distributions in the Peruvian upwelling, where diatoms dominate the algal assemblages. DMS flux densities, computed with seawater and atmospheric DMS measurements, indicated that the

Peruvian upwelling region was not a pronounced source of DMS to the atmosphere in both December 2012 and October 2015. Further attention should be given to the improvement of the understanding of DMS cycling as well as its underlying production





and consumption processes (e.g., regular monitoring) since our study reveals highly variable season/interannual DMS concentrations in the Peru upwelling system.

*Data availability*. DMS data for the M91 and SO243 cruises have been submitted to the NOAA PMEL database (http://saga.pmel.noaa.gov/dms). Phytoplankton pigment data for M91 and SO243 are available at PANGAEA at https://doi.pangaea.de/10.1594/PANGAEA.864786 and https://doi.pangaea.de/10.1594/PANGAEA.898920, respectively. For further data, please contact the corresponding author (yzhao@geomar.de).

*Author contributions*. YZ, DB, CAM, CS and HWB designed the study. DB, CAM, CS, AB and HWB participated in the fieldwork. Sulphur compound measurements and data processing were done by DB and CS. YZ conducted further data analysis and wrote the article with contributions from all co-authors.

*Competing interests*. The authors declare that they have no conflict of interest.


*Acknowledgements*. We thank the captain and crew of the R/V Meteor during cruise M91 and R/V Sonne during cruise SO243, as well as the co-chief scientist Damian Grundle (SO243). We would like to thank Patrick Veres and Bettina Derstroff for the atmospheric DMS measurements during M91. We would like to thank Kerstin Nachtigall for nutrient measurements (M91), Sonja Wiegmann for HPLC pigment analysis (M91, SO243) and Martina Lohmann for nutrient

measurements (SO243). We also thank Dr. Philip Froelich for providing data from the cruise the R/V Robert Conrad in 1982. The authors gratefully acknowledge NASA for providing the MODIS-Aqua satellite data as well as  NOAA Air Resources Laboratory (ARL) for the provision of the HYSPLIT model used in this publication. The cruise M91 was part of the German research projects SOPRAN II (grant no. FKZ 03F0611A) and III (grant no. FKZ 03F0662A) funded by the Bundesministerium für Bildung und Forschung (BMBF). The cruise SO243 was financed by the BMBF through grant

03G0243A. Astrid Bracher's contribution was supported by the Helmholtz Infrastructure Initiative FRAM. Elliot Atlas acknowledges support from the NASA Upper Atmosphere Research Program Grants NNX13AH20G and NNX17AE43G. Yanan Zhao is grateful to the China Scholarship Council (CSC) for providing financial support (File No. 201606330066).

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

525



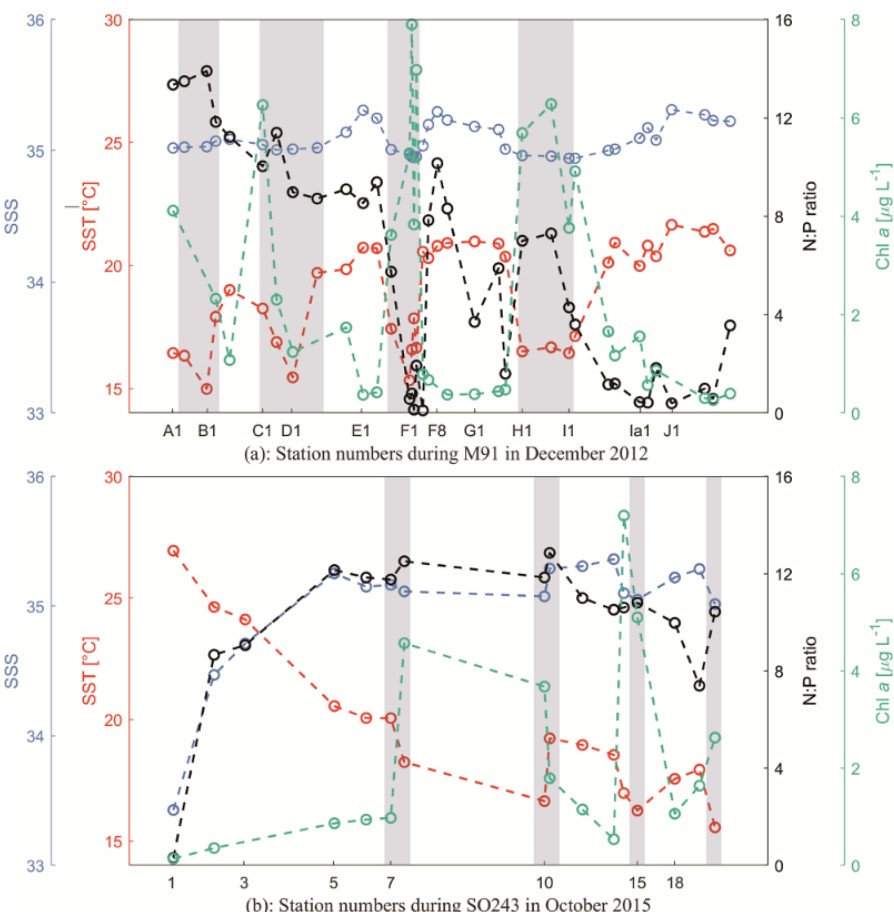

**Figure 2. Sea surface temperature (SST; red), Sea surface salinity (SSS; blue), N:P ratio (black; N stands for the sum of dissolved nitrate and nitrite, and P stands for phosphate) and Chl a (green) at each station during M91 (a) and SO243 (b). Grey rectangles indicate coastal CTD stations. All values from both cruises are integrated between 1–10 m.**







**Figure 3.** Surface seawater measurements of sulfur compounds are shown in panels a–f (circles) for $DMSP_p$ (M91), $DMSP_d$ (M91), $DMSO_p$ (M91), $DMSO_d$ (M91), DMS (M91) and DMS (SO243), respectively. Monthly (December and October) surface DMS concentrations from the Lana et al. (2011) climatology are colour-coded in the background in e and f, respectively. Diamonds in panel e, as well as diamonds and triangles in panel f, are recorded DMS concentrations in adjacent waters retrieved from the PMEL database for June–July 1982 (Andreae, 1985), October–November 2007 (unpublished DMS data) and October–November 2008 (Hind et al., 2011), respectively. Note that to ensure readability of the plots, the scale in panel e is capped at 8 nmol $L^{-1}$ with a few values exceeding the upper threshold, and DMS data retrieved from the PMEL database were averaged every ten samples.



**Table 1. Spearman's rank coefficients of correlations of all sulfur compounds surface data with selected abiotic and biological parameters. Bold numbers indicate correlations that are significant ($p < 0.05$) with a sample size of 27 for M91, 36 (environmental and biological parameters) and 13 (nutrients) for SO243 between 9–16 °S. ß-caro represents ß-carotene, diat represents diatoms, hapto represents haptophytes, and chloro represents chlorophytes. NA represents not available.**

| M91 | SST | SSS | N:P | Chl $a$ | ß-caro | Diat | Hapto | Chloro | DMSO$_d$ | DMSO$_p$ | DMSP$_d$ | DMSP$_p$ |
|---|---|---|---|---|---|---|---|---|---|---|---|---|
| DMS | 0.05 | -0.08 | **-0.51** | 0.11 | 0.12 | 0.29 | -0.20 | 0.37 | **0.68** | 0.28 | 0.34 | 0.09 |
| DMSP$_p$ | -0.15 | **-0.43** | **-0.40** | 0.28 | **0.39** | 0.25 | -0.14 | -0.18 | **0.56** | **0.92** | **0.71** | |
| DMSP$_d$ | 0.25 | -0.05 | **-0.46** | -0.13 | -0.01 | 0 | -0.08 | -0.01 | **0.90** | **0.50** | | |
| DMSO$_p$ | -0.11 | **-0.49** | **-0.63** | **0.39** | **0.46** | 0.35 | 0.10 | -0.17 | 0.31 | | | |
| DMSO$_d$ | 0.36 | 0.14 | **-0.52** | -0.23 | -0.14 | -0.07 | 0.04 | 0.03 | | | | |
| SO243 | | | | | | | | | | | | |
| DMS | -0.07 | -0.09 | **-0.61** | 0.01 | NA | 0.21 | -0.03 | -0.26 | | | | |

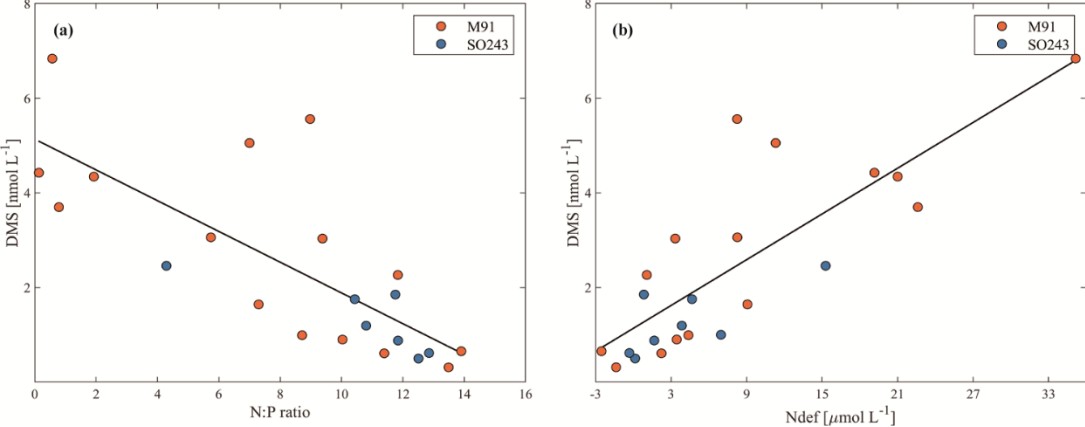

**Figure 4: (a): N:P ratio vs. surface DMS concentrations in the coastal Peruvian upwelling (y = -0.32x + 5.14, $r^2 = 0.58$, $p = 3.61 \times 10^{-5}$, n = 22). (b): Ndef vs. surface DMS concentrations in the coastal Peruvian upwelling (y = 0.16x + 1.14, $r^2 = 0.67$, $p = 3.62 \times 10^{-6}$, n = 22).**





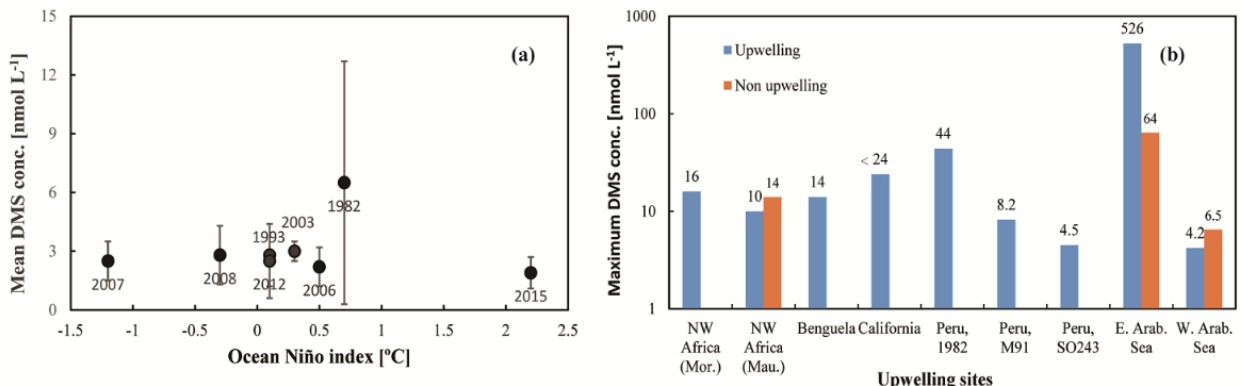

**Figure 5. (a): Mean and standard deviation of DMS concentrations for each cruise in the ETSP. Note that the value from 2003 only includes the data east of 95 °W. (b): Maximum DMS concentrations measured in various upwelling regions worldwide. Mor. stands for Morocco and Mau. stands for Mauritania.**

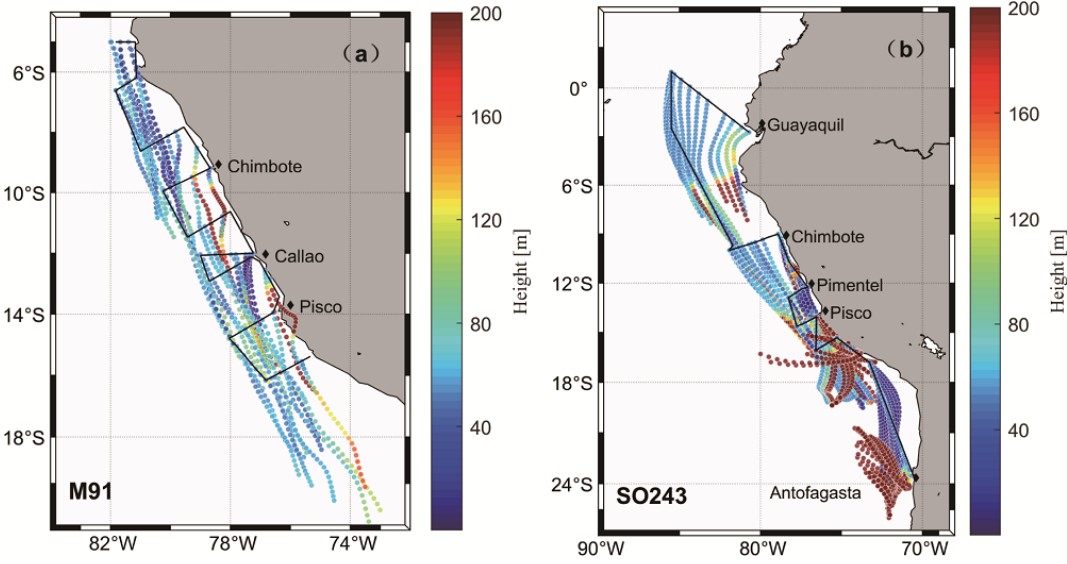

**Figure 6: Air mass backward trajectories of cruise M91(a) and SO243 (b), with colorbar indicating the height above sea level. Note that both scales are capped at 200 m to ensure readability of the plots, despite that some values exceed the upper threshold.**

565

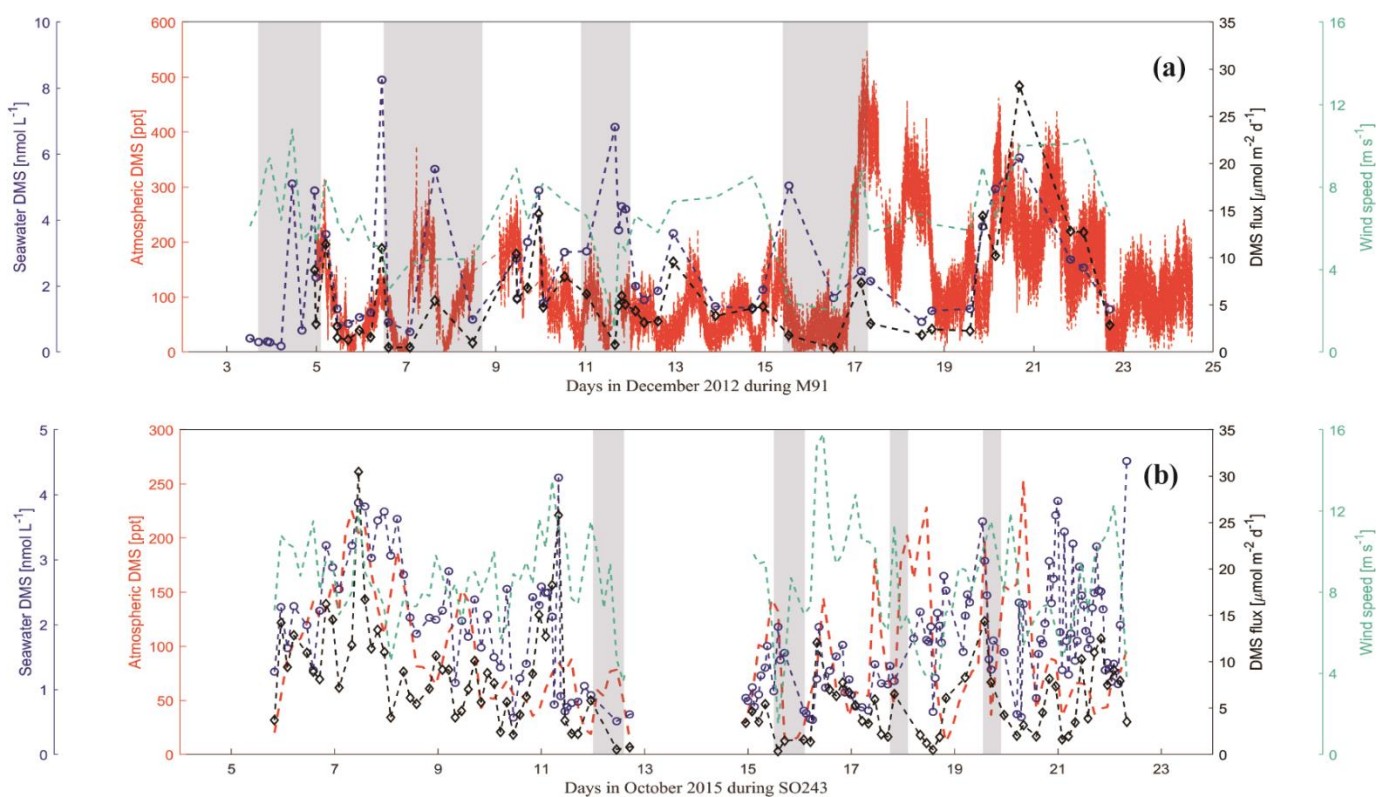

**Figure 7: Atmospheric DMS mole fractions (red dash line), surface seawater DMS concentrations (blue circles) and DMS fluxes (black diamonds) and wind speeds (green dash line) during the cruise M91 (a) and SO243 (b). Grey rectangles (a and b) indicate coastal CTD stations for both cruises. Note that DMS mole fractions during M91 were averaged for 10 min to calculate flux densities.**

570