# Peer review of "Dimethylated sulfur compounds in the Peruvian upwelling system"

_Biogeosciences, 2021_

## Author Comment (AC1)

Dear Byron Blomquist:

Thank you for your constructive and helpful comments which led us to reconsider our findings and the way they are presented. Below are our detailed line-by-line responses.

**General comments:**

**R1**: Line 17, 25-28 and elsewhere: The conclusion that seawater DMS was 'relatively low' seems too vague and imprecise. Better to just state the mean/range/variance of the observations and compare this to previous measurements. I wouldn't stress a comparison with Lana et al. 2011 too much (PMEL database is better). As you have mentioned, there isn't much data from the "Humboldt Current Coastal' province, as defined in Lana et al., and their seasonal extrapolation for this province from limited data is just a best guess. This is a region where we expect seawater DMS concentration to vary quite a bit, spatially and temporally (on seasonal, yearly and perhaps decadal timescales). Combined with previous measurements, these data provide a better picture of seawater DMS in this region than the Lana et al. gridded product.

Authors: We thank the referee for his suggestion, and we have modified the corresponding parts of the text, in order to present our results in a precise manner.

**R2**: Since this study doesn't represent extremes in the ENSO cycle it would be interesting to know how well previous studies have sampled ENSO variability. Can the authors discuss their study and previous measurements from the ENSO perspective a bit more? Are we still far from a representative sample of DMS variability during ENSO extremes? Should this be a focus of future studies?

Authors: We thank the referee for his questions on the ENSO topic. Initially, this work was designed to explore the effect of ENSO on the biogenic production of DMS. In addition to our DMS data, we also extracted all previous DMS data collected from the Eastern Tropical South Pacific (ETSP) from the PMEL database. However, we could not draw a definitive conclusion on whether ENSO would affect DMS concentrations, which was mainly attributed to the fact that no cruise took place in the middle of ENSO events. For example, the upwelling waters were still observed, at least during part of the cruises in 1982 and 2015, when two extreme EI Niño events were occurring. This indicates that they were going through developing EI Niño events, and as a result, they are not adequately representative of fully developed EI Niño events. For other previous DMS data, they were generally collected in open ocean areas in the ETSP, and they were less influenced by ENSO events (Bates and Quinn, 1997). Therefore, our understanding is still hindered by the limited DMS data set, especially in the Peruvian coastal regions, and that is also why we shifted from the original idea to the current version with only briefly describing the story of ENSO and DMS. Also, in addition to the ENSO events, the seasonal cycles of DMS along the Peruvian upwelling make the comparison between previous measurements and our measurements more complex. Overall, to justify the influence of ENSO on DMS and to gain a more comprehensive understanding, we think more data are required, especially those collected from fully developed EI Niño events. It is clear that the link between ENSO and DMS should be a focus of future studies as this will expand our knowledge, for example, on how DMS fluxes change during/after the EI Niño/La Niña events, and if they have a

significant impact on the regional/global climate.

**R3**: line 19 and elsewhere: the terminology' flux density' is a bit odd and not typically used in the air-sea flux community. It's potentially confused with usage like 'spectral density' for power spectra, etc. Better to just say 'flux' or 'fluxes'.

Authors: We have replaced all 'flux density' and 'flux densities' with 'flux' and 'fluxes', respectively.

**R4**: lines 129-131: Although it doesn't make much difference to the conclusions of your study, we should really discourage the use of transfer models like Nightingale 2000 for DMS flux estimates. Numerous direct studies of DMS air sea flux have been conducted over the past decade or more and the relationship with wind speed (or friction velocity) is closer to linear, especially over the wind speed range of your study. Transfer models based on highly insoluble tracers don't represent DMS transfer well (see Fig 3 in Bell et al. 2013). Better to use a model that's actually validated with direct DMS flux observations. The COARE gas transfer model has been used (e.g. Bell et. al. 2013, Blomquist et al. 2017) but for this paper a simple linear relationship is probably fine: e.g. Huebert et al. 2010, Fig 3 or Blomquist et al. 2017 Fig 5.

Authors: We thank the referee for this advice. We acknowledge that the DMS gas transfer coefficient exhibited more of a linear (as referee suggested) instead of a quadratic (e.g., Nightingale 2000) dependence with the wind speed. However, to allow a direct comparison with Lana's climatology, we decided to use the same equation which was adopted in Lana et al. (2011) to reduce the uncertainty in the comparison. In the future studies, we will surely avoid using transfer models like Nightingale 2000 for estimating DMS fluxes, but for this manuscript we will continue with N2000 as the overall conclusions are not changed. Please note, we did refer to the direct DMS covariance flux work of Marandino et al. (2009) and Yang et al. (2011), who also found a linear k-U dependence with measurements in the study region.

**R5**: I'm confused by the reference to 'terrestrial DMS sources'. What are these potential sources? I'm not aware of any, especially in the arid coastal climate of Peru and Chile. A more likely source of high atmospheric DMS variability would be hotspots close to shore, which might be implicated if trajectories in the marine boundary layer follow the coastline for some distance.

Authors: The referee is right, there are no references reporting atmospheric DMS directly emitted over the terrestrial regions of Peru and Chile. However, there are some references reporting that terrestrial sites could be a source of atmospheric DMS in South American, such as coastal marshes (Crutzen et al., 2000) and tropical soils (Jardine et al., 2015). Some references also reported terrestrial sources in other regions, such as biomass burning in Australia (Meinardi, 2003) and the Pearl River Delta in China (Chan et al., 2006). We have added these references to the corresponding text.

**Technical comments:**

line 15: don't need a comma after 'present'.

Authors: Done.

---

## Author Comment (AC2)

We thank Referee 2 for all comments on our manuscript. Please find below our answers to all of them.

**General comments:**

**R1**: In the manuscript, you note that there is a good correlation between the two employed techniques but that the results are not on a 1:1 line. I would recommend discussing a bit further for example mentioning that the PTR measurement is showing higher concentrations. Besides, while the PTR directly measures the DMS concentration in air there might be potential losses of DMS in the other technique. Was there a possibility to do a comparison in the lab of the two techniques? This could also help to find out if the instruments actually worked well. It would also help to know which E/N ratio was used for the PTR as that has an influence on the performance of the instrument and if it was calibrated before and after the measurements. Which m/z was used to track the DMS signal in the PTR?

Authors: We thank the referee for these questions. The referee is right, we did not mention which instrument showed higher concentrations, and we have added this information to the manuscript. We acknowledge that there might be some potential reasons for the discrepancy between the two instruments. For instance, the canisters can experience loss during storage, leading to lower values, and the PTR cannot distinguish between compounds at the same mass, leading to potentially higher values. In addition, it is possible that some issues occurred with the calibration and standardization which applied in one or both techniques and, therefore, led to the discrepancy between the two instruments. The latter is the most likely explanation, as the loss/artifact explanation would be unlikely to produce the good correlation between the methods. However, the exact reasoning cannot be addressed at this late stage. We have added the mentioned information to the supplemental material. The two instruments were not directly compared, as they were independent measurements made at different locations (PTR on board, canister filled on board and measured in the lab). The PTR-ToF-MS was operated under standard conditions, pressure 2.2 mb, E/N 137, and mass resolution between 3700-4000. The PTR-ToF-MS was calibrated at the beginning, during, and at the end of the cruise. DMS was measured at mass 63.026 and calibrated to a gravimetrically prepared pressurized standard (Apel Riemer Environ. Inc, USA). We have added this information to the text.

**R2**: Regarding the atmospheric DMS concentrations, I was wondering if you could compare your data to results from satellite data?

Authors: It is not clear what is meant by this comment. To the best of our knowledge, there are no direct measurements of atmospheric DMS by satellite. There are indeed estimates of dissolved DMS concentrations by using satellite data. This latter approach is, however, beyond the scope of our manuscript.

**R3**: I was wondering about the roughness of the sea during the cruises. For concentrations of gas and particulate species in the marine atmosphere, the structure of the surface of the ocean is quite important. Figure 7 shows wind speed that as is mentioned in the manuscript is often correlated with

DMS concentrations. What about wave breaking? Do you have measurements of wave height or where the waves started breaking? Was there wave breaking in the regions defined as "coastal stations"?

Authors: We agree that wave breaking plays an important role in the sea-air gas transfer process. Unfortunately, no measurements of wave height or wave breaking were made during the cruises. Additionally, the wind speed is shown and discussed (mostly) because typically used bulk flux calculations employ gas transfer parameterizations based on wind speed. Thus, wind speed is directly used to determine the fluxes. To date, there are no heavily used (vetted) gas transfer parameterizations using wave parameters.

**R4**: you mention "terrestrial DMS sources" – did you mean terrestrial sulfate sources? Or what would terrestrial DMS sources be?

Authors: Please see our comment to referee 1 (R5).

**Technical comments:**

Sometimes there is a strange thing with the font. For example on page 3, line 76: the symbols in "°S" are too close to each other, a similar issue occurs with the word "Niño" on page 3, line 91 and in later occurrences of the word. Page 2, line 43: in the sentence starting with "Some studies.." there is only 1 reference. Either change to "The study by xx .. "or add more references.

Authors: All corrected.

---

## Author Comment (AC3)

We thank Referee 3 for the detailed comments and have refined the manuscript as suggested.

**General comments:**

**R1**: Line 17-18 Why did not the authors take samples of DMSP and DMSO in October 2015?

Authors: We collected $DMSP_t$ and $DMSO_t$ samples in October 2015. However, the subsequent measurements showed unreliable results and, therefore, we decided to exclude these results from the discussion.

**R2**: Line 158-160 "The N:P ratio, defined as the ratio of the sum of nitrate ($NO_3^-$) and nitrite ($NO_2^-$) to dissolved phosphate ($PO_4^{3-}$) for both cruises, is a good indicator of nutritional status: high/low N:P ratios indicate nitrogen repletion/limitation." Why didn't authors consider ammonium ($NH_4^+$) which was also an important dissolved inorganic nitrogen? Would it make a significant effect on the conclusions if considering the concentration of $NH_4^+$?

Authors: Unfortunately, no $NH_4^+$ measurements were made during both cruises. The autoanalyzer used during M91 and SO243 had no $NH_4^+$ channel, and because of the well-known contamination problems, we decided not to apply the manual method for $NH_4^+$ measurements. In fact, the N:P ratios won't change much when including $NH_4^+$ because $NH_4^+$ are usually very low in the oxic surface layer. Additionally, there is no reference reporting the effect of $NH_4^+$ on the DMS.

**R3**: Line 167-168 "with the most abundant phytoplankton groups being diatoms (45 %), haptophytes (24 %), and chlorophytes (18 %) (2018)". "(2018)" here needs a reference.

Authors: Done.

**R4**: Line 168-169 "N:P ratios were generally between 8-13 in the Peru upwelling region during SO243, indicating slightly limiting nitrogen conditions." How to define the nitrogen repletion/limitation, what's the N:P ratio range?

Authors: We thank the referee for this question. We used the constant 12.6, which is the empirically determined N:P ratio of organic matter produced in productive waters, as the threshold for nitrogen repletion/limitation (Codispoti and Packard, 1980). We added this information to the manuscript.

**R5**: Line 202-203 "In contrast to our observations, Zindler et al. (2012) reported a general decreasing trend of DMSPt concentrations with decreasing N:P ratios (1-12). This may be because the response to nitrogen limitation differs among specific algae groups." What are the dominant algal groups in Zinder et al.'s (2012) study?

Authors: The dominant algal group were diatoms in Zinder et al.'s (2012) study and we added this information to the text.

**R6**: Line 205-207 "variability at the species or genus level might result in different responses under

nitrogen limitation." Here a reference is needed.

Authors: Done.

**R7**: 227-228 "Generally, both N:P ratios and Ndef significantly correlated with coastal DMS values in the surface waters". Did the DMS values and the concentrations of solo nutrients exhibit any relationship?

Authors: We thank the referee for this question. Only coastal DMS values from M91 negatively correlated with $NO_3^-$ ($r^2 = 0.34$, $p < 0.01$), while this is still lower than the correlation between DMS and N:P ratios ($r^2 = 0.50$ $p < 0.01$) during M91. Coastal DMS did not exhibit a relationship with any solo nutrients during SO243 ($NO_3^-$, $NO_2^-$ and $PO_4^{3-}$), and this is also the case when combining the two cruises together. Therefore, we believe that N:P ratios or Ndef provided additional information for the relationship between the nutrient availability and DMS in the Peru upwelling.

**R8**: Line 255-256 "A comparison with DMS data from other EBUS and the Arabian Sea illustrates that DMS concentrations off Peru (up to 44 nmol L-1) are higher". Why did not the authors discuss the difference in DMS between Peru 1982 and this study? The DMS concentrations in this study were significantly lower.

Authors: The comparison between Peru 1982 and this study was originally described in lines 236-242 in the manuscript. Now we have readjusted the order of the corresponding text in order to make it more logical (241-248).

**R9**: Line 279 "which might be a result of the discrepancy for DMS flux densities between the two cruises". The explanation was out of place. Don't put the cart before the horse. The discrepancy for DMS flux densities was influenced by the DMS concentrations in seawater and atmosphere and wind speeds. Therefore, the difference in the DMS concentration in the atmosphere could not be attributed to the discrepancy for DMS flux densities. Please explain it reasonably.

Authors: The referee is right. We have rephrased the statement.

**R10**: Line 289-290 "On average, the mean DMS flux densities in the first and third regions during SO243 are slightly higher/comparable to 5.5 $\mu mol\ m^{-2}\ d^{-1}$ reported from Marandino et al. (2009)" Where did the first and third regions represent? Please define them.

Authors: We have rephrased the statement in the manuscript.

**R11**: Line 342-Line 343 and Line 358-360. Please check the references carefully.

Authors: Corrected.

---

## Referee Report (RR1)

I think the authors have addressed the most significant concerns and this version is suitable for publication with a couple minor additions. If they wish to use the Nightingale transfer coefficient model to estimate DMS fluxes for comparison with Lana et al., that's OK, but they should indicate that we now know the models with quadratic wind speed dependence are not representative of DMS transfer and will give significant overestimates for wind speeds above 10 m/s.

Food for thought: A revision to the Lana et al. 2011 global DMS model is in preparation. I am hoping authors of this revision will address some of the deficiencies in the DMS transfer estimate. But, direct observations from the region covered by this manuscript are still sparse and not representative of the entire ENSO cycle (according to the conclusions of this paper). Nutrient measurements from this region are likely much better represented over all seasons. I wonder if the authors feel the correlation with N:P could be used to help validate the reasonableness of the new global DMS climatology, or perhaps even find use as a predictive factor for surface seawater DMS?

---

## Author Response (AR2)

**Author's response:**

We thank Byron Blomquist for his suggestions. We have prepared a revised version of the manuscript based on these comments and highlighted these changes in yellow. Below are our detailed line-by-line responses.

**R1**: I think the authors have addressed the most significant concerns and this version is suitable for publication with a couple minor additions. If they wish to use the Nightingale transfer coefficient model to estimate DMS fluxes for comparison with Lana et al., that's OK, but they should indicate that we now know the models with quadratic wind speed dependence are not representative of DMS transfer and will give significant overestimates for wind speeds above 10 m/s.

Authors: The referee is right. We have added this information regarding the transfer models in the manuscript (Section 2.3).

**R2:** Food for thought: A revision to the Lana et al. 2011 global DMS model is in preparation. I am hoping authors of this revision will address some of the deficiencies in the DMS transfer estimate. But, direct observations from the region covered by this manuscript are still sparse and not representative of the entire ENSO cycle (according to the conclusions of this paper). Nutrient measurements from this region are likely much better represented over all seasons. I wonder if the authors feel the correlation with N:P could be used to help validate the reasonableness of the new global DMS climatology, or perhaps even find use as a predictive factor for surface seawater DMS?

Authors: We tried to incorporate the N:P ratios and corresponding DMS concentrations which were measured previously in surface seawater off the Peruvian upwelling in June/July 1982 (Andreae, 1985) into our analysis. However, the N:P ratios and DMS concentrations from Andreae (1985) were not correlated, which indicates that N:P ratios are not a good predictor for surface seawater DMS over all seasons off the coastal Peruvian upwelling. Therefore, we think N:P values cannot help with the validation or prediction of DMS in this revised climatology.